# Prediction of Optimized Color Design for Sports Shoes Using an Artificial Neural Network and Genetic Algorithm

**Yu-En Yeh**

Department of Multimedia Animation Design, TransWorld University, No. 1221, Zhennan Road, Douliu City, Yunlin County 640, Taiwan; yuen.yeh@msa.hinet.net

**Abstract:** Product design is a complicated activity that is highly reliant on individual impressions, feelings and emotions. Back-propagated neural networks have already been applied in Kansei engineering to solve difficult design problems. However, artificial neural networks (ANNs) have a slow rate of convergence, and find it difficult to devise a suitable network structure and find the global optimal solution. This study developed an ANN-based predictive model enhanced with a genetic algorithm (GA) optimization technique to search for close-to-optimal sports shoe color schemes for a given product image. The design factors of the sports shoe were set as the network inputs, and the Kansei objective value was the output of the GA-based ANN model. The results show that a model built with three hidden layers ($28 \times 38 \times 19$) could predict the object value reliably. The $R^2$ of the preference objective was equal to 0.834, suggesting that the developed model is a feasible and efficient tool for predicting the objective value of product images. This study also found that the prediction accuracy for shoes with two colors was higher than that for shoes with only one color. In addition, the prediction accuracy for shoes with a relatively familiar shape was also higher. However, the prediction of color preferences is relatively difficult, because the respondents had different individual color preferences. Exploring the sensitivity and importance of the visual factors (form, color, texture) for various image words is a worthy topic for future research in this field.

**Keywords:** product color design; Kansei engineering; neural network; genetic algorithm; sports shoes

## 1. Introduction

Within the current highly competitive marketplace, how to induce emotional resonance is crucial to a successful product design. A product's appearance is evaluated according to its overall image. A product with a favorable appearance requires the selection of the optimal combination of various design factors, such as form, color, texture, interface and line elements. The form of a product is defined by the relationships between a product's style, expression and function; modifying or emphasizing a product's volume or shape can attract the attention of consumers [1]. Additionally, an effective method is required to help designers present their design concept and elicit responses from consumers [2,3].

Color has a strong effect on us, beginning in childhood, and it can have highly emotional and symbolic associations when used in fashion, and even when used to emphasize form. Some studies have demonstrated that product color has a greater effect on the consumer perception of a product than the product form [4]. Color planning has been extensively studied, and has been adopted in product design [5–8]. For consumers evaluating an overall product image, color and form are mutually dependent.

If designers could follow a design guideline or a reliable information system to match a product with a color (or colors), based on the information from a large-scale survey, then they could achieve close to an optimal color design for customized product design services.

Sports shoes are a classic example of the mass customization of popular products. Recent studies have investigated variations in the manufacture of shoes. For instance, Huang proposed a size recommendation framework based on both 3D (foot and last) features and user preferences and provided a predictive model of the comfort levels for particular parts of the foot on the basis of the given size recommendation [9]. Lee and Wang classified the foot shapes of Taiwanese people using 3D foot scanning data; they asserted that women have a longer ball of foot length than do men (0.2% FL), and that when comparing feet of the same length, men have a longer breadth, girth and height, but a shorter toe height than do women [10]. Alcantara et al. used principal component analysis and factor analysis to determine consumer preferences and perceptions of casual footwear [11]. Chang et al. applied fuzzy set concepts to build a jogging shoe prototype expert system [12]. Sudta et al. combined a decision tree model, a k-nearest neighbor (KNN) model, and a neural network (NN) model, to present a prototype web application for providing suggestions for children's shoes [13]. Taken together, these findings provide useful information for shoe development.

In the author's previous study, partial least squares (PLS) and neural networks were introduced to explore shoe form design factors and consumers' image perceptions [14]. In a subsequent study, the author used Kansei engineering steps, semantic differential rating, and statistical tools, to investigate the consumer psychology, perception and aesthetics in relation to sports shoe colors [15].

To follow up, this study presents an in-depth hybrid neurogenetic optimization methodology based on ANNs and GAs to build a product color design prediction framework for shoe designers, by integrating customers' feelings regarding form and color in shoe samples. The NN is used to learn the experimental values obtained from experimental samples using Kansei engineering. Sports shoe design factors are used as the input parameters of the NN.

The nonlinear relationship between consumers' perceptions of a shoe's image and its color scheme is determined by the NN, whereas the GA is employed to search the optimized NN's architectural parameters.

## 2. Literature Review

### 2.1. Kansei Engineering

Kansei engineering was developed by Nagamachi in the 1970s [16]. It uses a questionnaire to link a customer's feelings about a product with image words, and transform consumer emotions into actual design elements. The Kansei concept has been used widely in subsequent decades, and in the design of product form and color [17–19].

Numerous quantitative analysis tools are used in Kansei engineering; these include multiple regression analysis [20], quantification theory type 1 [21], fuzzy theory logic [22], rough set theory [23], procrustes analysis [24], genetic algorithms (GAs) [25] and neural networks (NNs) [26]. However, modeling using ANNs causes some difficulties for the construction of reliable Kansei predictive models regarding the number of hidden layers and the learning parameter settings. To overcome these limitations, ANNs can be merged or hybridized with other techniques, such as fuzzy logic, wavelet transformation, partial least squares (PLS), rough sets and GAs [27].

Among these, GA solves the optimization problem by mimicking processes in biological evolution; GAs are also cost-effective and less time consuming than other approaches. GA thus provides a versatile problem-solving mechanism for searching, adaptation and learning in a variety of application fields, and especially for those problems that are solved unsatisfactorily by heuristic methods. Therefore, a Gam that is assisted by NN can be an effective tool for predicting and optimizing complex process parameters.

## 2.2. ANNs

Neural network technology was developed by Gallant in 1993 [28,29]. The main features of an ANN are that it can mimic the self-learning and organization ability of the human brain, it is capable of handling incomplete data, and can solve complex and ill-defined problems. The basic computational units of the neural network are referred to as nodes, and nodes are connected in three layers: the input, output and hidden layers. The hidden layer extracts important features contained in the input data [28]. In Kansei engineering, ANNs have been used to determine relationships between product design attributes, such as size, shape and color, and also identify brand and consumer emotional responses in order to make Kansei predictions.

A few studies have illustrated the use of ANNs in the product design field. Tsai, Hsiao and Hung proposed a conceptual design method that used fuzzy ANN-based algorithms, integrating product form and color to predict consumers' evaluation of the product's image [30]. Recent study regarding sports-shoe appearances was published by the author of this study, PLS partial least squares and ANN were used to developing a NN-based Kansei prediction system [31]. Lai used ANNs to determine the optimal combination of product form and product color [32]. An ANN model was also used to examine the complex relationship between web page design elements and users' feelings, ultimately building a web page design support database [33]. Tang et al. established an ANN model to relate the design parameters of a new product to its perceptual value, and used a mobile phone design as a case study [34].

Once trained, an ANN can perform predictions and generalizations at high speeds, and the root-mean-square error (RMSE) is used to analyze model prediction accuracies [35]. Compared with linear modeling techniques, such as multilinear regression and PLS, ANNs are superior as a modeling technique for molecular descriptor data sets showing nonlinear conjunction, and thus for both data fitting and prediction strengths [36]. Among the different learning algorithms, backpropagation uses the concept from the method of steepest descent, a gradient method, to minimize the error function; this makes backpropagation the most common way of training ANNs. The industrial design applications of ANNs typically focus on feature selection, classification and product image prediction. They can manage large-scale investigations of nonlinear data, such as product design attributes or consumer emotional responses to provide consumer Kansei reaction predictions, sort and categorize product attributes, and perform grouping and regression [37].

However, ANNs have a number of shortcomings, such as their low learning rate [38] and difficulty determining the structure of hidden layers when devising a suitable network structure [39]. It is also not easy to achieve the global optimal solution [40], initialize the network's weights, or explain the training results.

## 2.3. GA

A GA is a stochastic search algorithm and optimization technique based on the concept of natural selection and the genetic evolution of biological species; it is also cost-effective and less time-consuming than other techniques. GAs search from a population of solutions rather than randomly from a single point. The GA optimization process can be terminated by defining some stopping criteria, such as the maximum number of generations, or the desired fitness being met. There are three basic operators in a GA: selection, crossover and mutation. Briefly, the algorithm starts by generating a random population of chromosomes, which are the candidate solutions to the given problem. Then, it evaluates the fitness function of each chromosome, which determines its probability in the selection stage. The crossover operation is applied to a pair of selected chromosomes, by combining them in order to generate a new, better, superior offspring [41]. Lampinen developed a GA approach for preliminary cam design and subsequent shape optimization [42]. Beale constructed a data mining model that used a GA to measure the correlation between web pages and user interests [43]. The inverse procedure is performed to generate a second offspring [44]. In recent years, GAs have been applied to solve optimization problems in numerous fields [45,46], including in design.

For instance, Beale constructed a data mining database that uses GAs to measure the correlation between web pages and user interests, Hsiao demonstrated the effectiveness of GAs for assessing coffee maker design feasibility [46], and Wang employed the interactive GAs and the fuzzy kano model to explore the emotional needs of users for electric bicycle design.

*2.4. Hybrid GA-Based ANN*

ANNs have many advantages in a variety of practical fields and applications; however, a slow rate of convergence and difficulty finding the global optimal solution are the major drawbacks of implementing ANNs. A GA assisted by an ANN can be an effective method of predicting and optimizing any complex process parameters for increasing the success of ANN modeling [47].

The main advantage of GAs is their ability to avoid becoming trapped in a local optimum [48,49]. GAs contribute to ANN model performance through the network weight initialization. Because ANN training algorithms are based upon local search procedures, they are prone to local minima problems. To avoid this, initial network weights that lead to a global minimum should be introduced, and in a hybrid, GA-based ANN model, the network connection weights and biases are not randomly generated but optimized using the GA [50]. While searching for the global solution, the GA is proposed to function on the trained neural network; the random nature of the GA helps the model to get away from local optima and avoid overfitting. Therefore, the combination of ANN and GA can be used for an integrated process of modeling and optimization.

The hybrid GA-based ANN technique was recently applied in the field of design. Hsiao used a fuzzy ANN to establish the relationships between input form design parameters and adjectival image description; they then use a GA to search for a near-optimal design that satisfies the designer's required product image. Tang et al. proposed a parametric approach that uses a three-layered ANN model, incorporated with a GA and the technique of generalized superellipse fitting for product aesthetic design [51]. Ming-Chyuan Lin et al. propose an integrated ANN, GA and the Taguchi quality design process to aid the search for the optimal solution with the most precise design parameters in order to improve product development [52]. The actual processes used to develop a hybrid GA-based model and GA optimization settings are discussed later in this paper.

## 3. Implementation Methods

*3.1. Selection of Representative Form Types and Color Schemes to Build the Experimental Sample Images*

When viewing and evaluating a product, consumers are affected by the interaction between the overall shape and color. Therefore, shape change was incorporated as a factor in this study that approximated actual product presentation and evaluation conditions to build a Kansei-based sport shoe color design prediction model. First, four types of currently-available sports shoes were selected from the current market based on their functionality and features: basketball, jogging, running and casual shoes. The final type of shoe was selected from the three-dimensional (3D) model resource, and was unmodified by product design considerations (a 3D model of an ordinary sports shoe that had not been produced through product design and aesthetic evaluation). These five form types were the experimental form factors in this study. Once the form of the five experimental shoe types was confirmed, each type was given two color schemes, with the shoe body in the main color and the outsole in the secondary color. Thus, a total of ten experimental color schemes were produced (Table 1).

**Table 1.** Color schemes of the experimental designs.

| Shoe Name | Form Type | Color Scheme 1 | Color Scheme 2 |
|---|---|---|---|
| Basketball shoes |  |  |  |
| Jogging shoes |  |  |  |
| Casual Shoes |  |  |  |
| Running Shoes |  |  |  |
| Unmodified |  |  |  |

### 3.2. Linguistic Variable Selection for Affective Responses

To define consumers' affective responses for the Kansei factor space, the semantic space was used to explore flow using the following steps. First, 100 antonymic pairs of Kansei expressive adjectives (in Chinese), that could be used to describe the sports shoes, were selected. Words that had identical or similar meanings, or that were semantically misleading, were excluded, with only 20 pairs of adjectives retained. Subsequently, 15 graduate students and four product designers were recruited to examine the 10 representative 3D-rendered images of sport shoes, looking at them in a random order. A seven-stage semantic differential (SD) questionnaire was designed and completed by the students. The collected data were summed and averaged, and the results were processed using factor analysis to determine the adjectives most representative of the sports shoes' color schemes. Finally, cluster analysis was employed to select three adjective pairs from the SD questionnaire data: elegant–artless, rare–common and like–dislike.

### 3.3. Definition of Experimental Color Index for Sports Shoes

The Practical Color Coordinate System (PCCS) was developed by the Japan Color Research Institute in 1964, and is based on psychological elements. The PCCS-based 24-color color wheel is displayed in Table 2. The main feature of the PCCS is its hue–tone system; thus, the PCCS has the advantage of treating color as an image by using its tone. The classification system is relatively similar

to the expression and description of colors in everyday life. Therefore, in the PCCS, tones are named according to investigation results, so that colors with the same hue can be distinguished by their tones. For example, in color names such as vivid red, light purple, soft orange and pale green, red, purple, orange and green are hues, whereas vivid, light, soft and pale denote tones. The 12 color tones are vivid (v), bright (b), strong (s), deep (dp), light (lt), soft (sf), dull (d), dark (dk), pale (p), light grayish (ltg), grayish (g) and dark grayish (dkg). The PCCS comprises red, yellow, green and blue, which are referred to as the four primary psychological colors, and are at the center of the color circle. In opposition to the four hues, four complementary psychological colors are identified, totaling eight main colors. Among the aforementioned eight hues, four hues are inserted at regular intervals to produce 12 hues. In addition to the 24 colors, ten gradient colors, plus black and white, two colors that are commonly seen in sports shoe design, are added to make a total of 36 colors. These colors were adopted as the main source of color changes for the experimental colors applied to sports shoe types in this study (Table 3).

**Table 2.** Structure of 24-color Practical Color Coordinate System (PCCS) color wheel.

| PCCS-based 24-color color wheel | Hue Code | Hue Name | Hue Code in the Munsell Color System | Hue Code | Hue Name | Hue Code in the Munsell Color System |
|---|---|---|---|---|---|---|
|  | 1:pR | purplish red | 10RP | 13:bG | bluish green | 9G |
| | 2:R | red | 4R | 14:BG | blue green | 5BG |
| | 3:yR | yellowish red | 7R | 15:BG | bluie green | 10BG |
| | 4:r0 | redish orange | 10R | 16:gB | greenish blue | 5B |
| | 5:0 | orange | 4YR | 17:B | Blue | 10B |
| | 6:y0 | yellowish orange | 8YR | 18:B | Blue | 3PB |
| | 7:rY | redish yellow | 2Y | 19:pB | purplish blue | 6PB |
| | 8:Y | yellow | 5Y | 20:V | Violet | 9PB |
| | 9:gY | greenish yellow | 8Y | 21:bP | bluish purple | 3P |
| | 10:YG | Yellow green | 3GY | 22:P | Purple | 7P |
| | 11:yG | yellowish green | 8GY | 23:rP | redish purple | 1RP |
| | 12:G | green | 3G | 24:RP | Red purple | 6RP |

Table 3. The 36 experimental color scheme designs.

| NO | Sample | R.G.B | NO | Sample | R.G.B | NO | Gradient Color |
|---|---|---|---|---|---|---|---|
| 1 | | 230.0.51 | 14 | | 0.160.193 | 27 | |
| 2 | | 229.0.79 | 15 | | 0.158.150 | 28 | |
| 3 | | 229.0.106 | 16 | | 0.155.107 | 29 | |
| 4 | | 228.0.127 | 17 | | 0.153.68 | 30 | |
| 5 | | 190.0.129 | 18 | | 34.172.56 | 31 | |
| 6 | | 146.7.131 | 19 | | 143.195.31 | 32 | |
| 7 | | 96.25.134 | 20 | | 207.0.219 | 33 | |
| 8 | | 29.32.136 | 21 | | 255.251.0 | 34 | |
| 9 | | 0.71.157 | 22 | | 252.200.0 | 35 | |
| 10 | | 0.104.183 | 23 | | 243.152.0 | 36 | |
| 11 | | 0.134.209 | 24 | | 235.97.0 | | |
| 12 | | 0.160.233 | 25 | | 230.0.18 | | |
| 13 | | | 26 | | | | |

### 3.4. Semantic Meaning Questionnaires to Determine Affective Responses

Once the five prototype sports shoe models had been chosen, each model was assigned two types of color scheme (Table 4). In addition, 36 representative color samples (Table 3) were used. A total of 360 3D-rendered sports shoe images were created, a selection of which are displayed in Table 4.

A total of 250 people participated in this study; 125 were men, and the remainder were women. All participants were college students and had no particular brand association. The questionnaires experiment used a cloud database, and was conducted by having the participants view the rendered sports shoe sample images. The images were shown in a random order on a fixed questionnaire interface (Figure 1). During the survey, the participants were instructed to make judgments based on their perceptions of the product in the image. The participants were required to rate the three rating pairs: Adj1 (elegant–artless), Adj2 (rare–common), and Adj3 (like–dislike) by using a scale from −10 to

10 (10 indicated the extremely positive impression of a sample, whereas −10 indicated an extremely negative impression). The evaluation data collected from the questionnaire were the input variables in the subsequent model construction, which is presented in the following section.

**Table 4.** Design factors and design factor levels for product forms of sports shoes.

| Design Factors | Design Factor Levels | | |
| --- | --- | --- | --- |
| $X_1$: Form type (Data type: discrete) | $X_{11}$ Basketball shoes<br>$X_{12}$ Jogging Shoes<br>$X_{13}$ Running Shoes<br>$X_{14}$ Casual Shoes<br>$X_{15}$ Other type | | |
| $X_2$: Number of colors (Data type: continuous) | $X_{21}$ Single Color<br>$X_{22}$ Two Colors | | |
| $X_3$: Direction of color gradient | $X_{31}$ Horizontal Gradient<br>$X_{32}$ Vertical Gradient | | |
| $X_4$: Sole color | $X_{41}$ White<br>$X_{42}$ Black | | |
| $X_5$: Experimental color index<br>Body color | Color index | $X_{51}, \ldots, X_{526}$ 26 Experimental color index | |
| | | $X_6$ Red (R) $X_7$ Green (G) $X_8$ Blue (B) | |

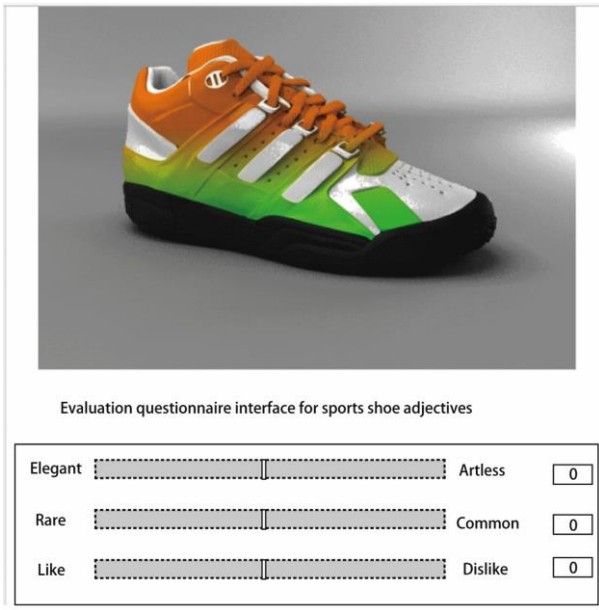

**Figure 1.** Interface of web-based questionnaire for adjective evaluation.

## 4. Results and Discussion

### 4.1. Definition of Design Factors for Modeling

The images of the representative sports shoes in this study had two types of design factor—form factors and color factors—which were also divided into sublevels (Table 5). For example, the number of colors $X_2$ was subdivided into two feature levels: single color ($X_{21}$) and two colors ($X_{22}$). The sole color $X_4$ was also subdivided into two feature levels: white ($X_{41}$) and black ($X_{42}$). The body color index was represented by $X_{51}$–$X_{526}$. In the data type and data values, the form presented must be observed to facilitate the numerical data format and conversion. Table 6 displays the coding format used for ANN training, and the average Kansei evaluation ratings of 12 selected sports shoe samples, presented in a matrix.

**Table 5.** Kansei evaluation matrix for 12 selected sports shoe samples.

| Model | $X_1$ | | | | | $X_2$ | | $X_3$ | | $X_4$ | | $X_5$ | $X_6$ | $X_7$ | Adj. 1 | | Adj. 2 | | Adj. 3 | |
|---|---|---|---|---|---|---|---|---|---|---|---|---|---|---|---|---|---|---|---|---|
| | $X_{11}$ | $X_{12}$ | $X_{13}$ | $X_{14}$ | $X_{15}$ | $X_{21}$ | $X_{22}$ | $X_{31}$ | $X_{32}$ | $X_{41}$ | $X_{42}$ | | | | Ave. | StdDev | Ave. | StdDev | Ave. | StdDev |
| typa_g01 | 1 | 0 | 0 | 0 | 0 | 0 | 1 | 1 | 0 | 0 | 1 | 255 | 251 | 0 | 0.14 | 3.11 | 0.75 | 2.96 | 1.81 | 3.03 |
| Typa01 | 1 | 0 | 0 | 0 | 0 | 1 | 0 | 0 | 0 | 0 | 1 | 230 | 0 | 18 | −0.83 | 3.13 | 1.10 | 2.26 | 0.52 | 3.18 |
| typb_g01 | 1 | 0 | 0 | 0 | 0 | 0 | 1 | 1 | 0 | 1 | 0 | 255 | 251 | 0 | −1.00 | 3.05 | 0.00 | 2.83 | 1.22 | 3.21 |
| typb01 | 1 | 0 | 0 | 0 | 0 | 1 | 0 | 0 | 0 | 1 | 0 | 230 | 0 | 18 | −1.46 | 2.44 | −0.21 | 2.15 | 0.00 | 2.45 |
| typc_g01 | 0 | 1 | 0 | 0 | 0 | 0 | 1 | 1 | 0 | 1 | 0 | 255 | 251 | 0 | −2.48 | 2.53 | −0.10 | 2.86 | −0.48 | 3.36 |
| typc01 | 0 | 1 | 0 | 0 | 0 | 1 | 0 | 0 | 0 | 1 | 0 | 230 | 0 | 18 | −1.11 | 2.13 | 0.61 | 1.81 | −0.25 | 2.46 |
| typd_g01 | 0 | 1 | 0 | 0 | 0 | 0 | 1 | 1 | 0 | 0 | 1 | 255 | 251 | 0 | −0.90 | 3.23 | −0.53 | 2.42 | 0.47 | 2.89 |
| typd01 | 0 | 1 | 0 | 0 | 0 | 1 | 0 | 0 | 0 | 0 | 1 | 230 | 0 | 18 | −1.39 | 3.32 | −0.19 | 2.99 | −0.55 | 3.77 |
| type_g01 | 0 | 0 | 1 | 0 | 0 | 0 | 1 | 1 | 0 | 1 | 0 | 255 | 251 | 0 | −1.58 | 2.98 | −1.00 | 2.89 | 0.58 | 3.30 |
| type01 | 0 | 0 | 1 | 0 | 0 | 1 | 0 | 0 | 0 | 1 | 0 | 230 | 0 | 18 | −2.03 | 2.80 | −0.76 | 2.72 | −0.07 | 3.23 |
| typf_g01 | 0 | 0 | 1 | 0 | 0 | 0 | 1 | 1 | 0 | 0 | 1 | 255 | 251 | 0 | −1.92 | 2.71 | −0.81 | 2.42 | 0.81 | 3.45 |
| typf01 | 0 | 0 | 1 | 0 | 0 | 1 | 0 | 0 | 0 | 0 | 1 | 230 | 0 | 18 | −1.29 | 2.31 | −1.07 | 2.05 | −0.39 | 2.45 |

**Table 6.** Genetic algorithm (GA) structural parameters.

| | |
|---|---|
| **Population size** | 100 |
| **Maximum number of generations** | 100 |
| **Termination criterion** | Meet the maximum number of generations (100) or tolerance ($10^{-6}$) |
| **Selection function** | Roulette wheel |
| **Crossover function** | Discrete method (0.65 probability of crossover) |
| **Mutation function** | Real valued (0.05 probability of mutation) |

*4.2. Description of the ANN Model*

Figure 2 illustrates the framework of the ANN model determined in this study. The framework is divided into input, hidden and output layers. The default type of transfer function is selected for hidden and output layers in MATLAB. The hyperbolic tangent sigmoid transfer function (tansig) for the hidden layer, and the log-sigmoid transfer function (logsig) for output layer. Backpropagation is used to train the ANN. The input layer is where the sports shoe design factors are input, and can be further divided into discrete and continuous factor types. The discrete factors can be further classified as form or color factors (Table 5). The hidden layer mainly serves to increase the complexity of the ANN and represents the interaction between inputs. The hidden layer is a structure describing the complex interactions among various variables in the ANN structure. The number of neurons and hidden layers cannot be known in advance. This study determined an appropriate number of hidden layers through trial and error. The output layer expressed the scores which the respondents gave for the perceived adjective-based scale anchors. ANN can be simulated using these data to establish a nonlinear relationship between shoe design factors and Kansei adjectives.

First, a 10 × 10 ANN structure was used to build the correlations between shoe design factors and adjectives in this experiment. Because the ANN structure could not be easily determined, and there were no specific sets of rules to follow, trial and error was used to decide the structure. Figure 3 plots the preliminary training results for the three sets of adjectives, where the horizontal axis indicates the experimental value, and the vertical axis indicates the predicted value. The $R^2$ for Adjs.1–3 were 0.536, 0.555 and 0.757, respectively. The results indicated, that except for Adj3 (like–dislike), which obtained relatively satisfactory prediction results through training, the other two adjective pairs [i.e., Adj1 (elegant–artless) and Adj2 (rare–common)], did not achieve satisfactory prediction results through training. Therefore, this study inferred that building an ANN framework with complicated relationships for the experimental factors might be relatively difficult. Using trial and error was not effective enough to rapidly identify valid model-building parameters and generate accurate predictions. Therefore, the GA was employed to find the optimal structure and build a more accurate ANN model.

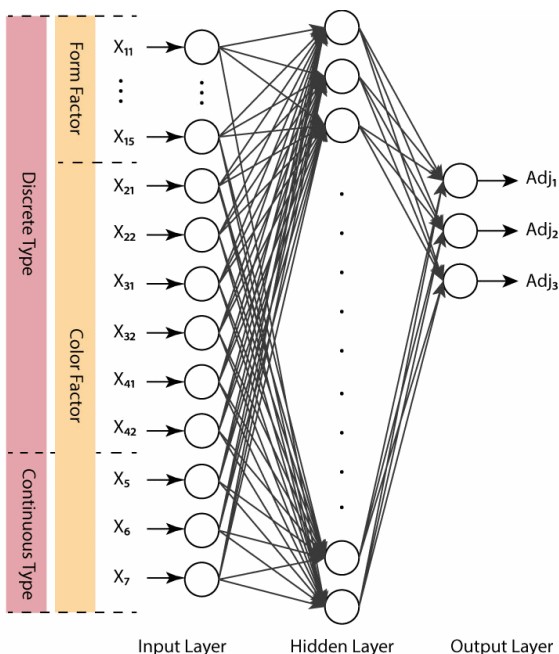

**Figure 2.** Artificial neural network (ANN) model structure.

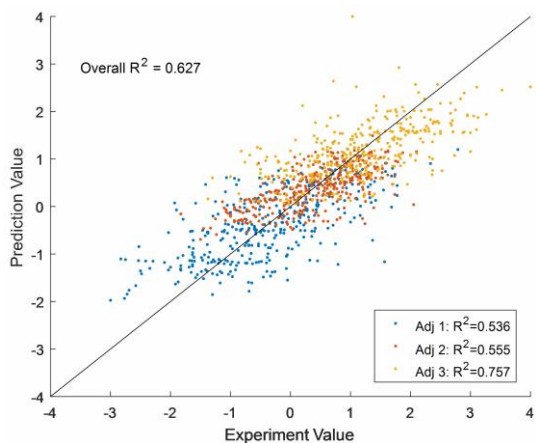

**Figure 3.** The $10 \times 10$ ANN structure training result for three adjective pairs.

### 4.3. GA Optimization Search Mechanism

This section details the use of the GA for the sports shoe color scheme optimization. The main components of the GA algorithm are chromosomes, population, generation and fitness value. In this study, the largest output value (adjective score) is used as the target to be obtained using genetic rules, whereas the chromosome represents the input variables, which are the shoes' design factors (binary data). The initial chromosomes in the first generation were generated at random. The fitness value was calculated, and the smaller the fitness value, the better the fit. The algorithm converges toward the optimal image as the RMSE decreases. In the next generation, new chromosomes are determined using reproduction, crossover and mutation operations. The new fitness was then calculated, and the favorable chromosomes were retained. The calculations proceeded until the maximum number of generations, or the tolerance in the difference between the old and new chromosomes, was reached, indicating the best chromosome. In this study, the MATLAB Toolbox settings were 100 generations, and a tolerance of $10^{-6}$. The detailed GA structural parameters are listed in Table 6.

### 4.4. Hybrid Learning of ANN

In the first step of the GA-based ANN learning, the questionnaire data underwent postprocessing, and were formatted as the input and output data sets. The input nodes of the network were encoded into chromosomes in terms of real number and binary coding with respect to their nature. During this stage, the ANN initial weights were set using GA, and the minimum error was reached using a backpropagation algorithm. This temporary network design was not necessarily expected to produce favorable ANN model results, because its primary purpose was to find the minimum error of the network with respect to the selected input variables. The next step was to determine the optimum number of ANN hidden nodes, with the lower and upper boundaries fixed at 1 and 50, respectively. During the next stage, the number of hidden layers was increased by 1, and the previous steps were repeated. A new solution was obtained by the GA, and the NN used it to determine its fitness value, and this fitness was compared with that of the previous model. These steps were repeated until the minimum error was reached. The overall procedure is illustrated in Figure 4.

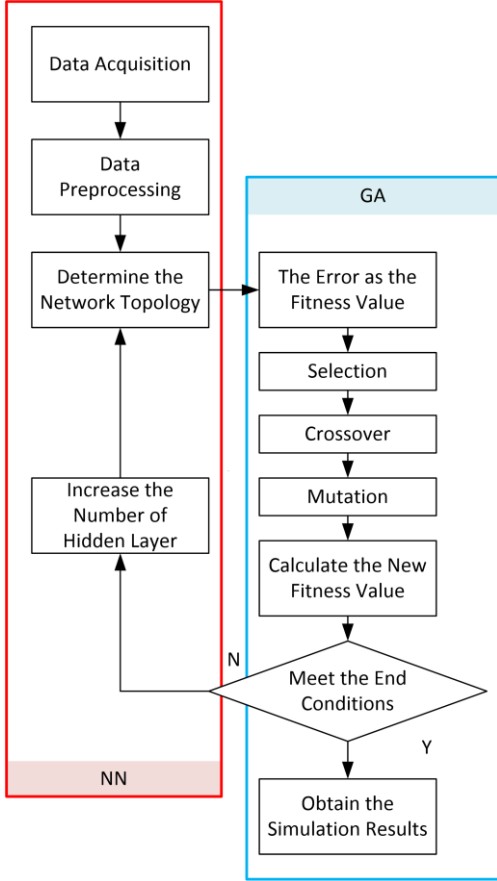

**Figure 4.** Flow chart of hybrid learning of GA-based ANN.

After the most relevant inputs and an optimal number of hidden nodes were fixed, the GA was used to initialize the weights of the network. At this stage, the GA selected initial weight vectors for each individual in the population, which represented the ANN initial weights. The individual weights were then trained using the Lavenberge–Marquardt algorithm, previously selected input variables, and an optimal number of hidden nodes until convergence was obtained. The GA provided initial values for the ANN that were near the global minimum. Therefore, the final solution was expected to fall into or near the global minimum.

Table 7 presents the performance of different GA-based ANN models, which were assessed using values of the mean squared error (MSE) and $R^2$. The models were composed of different numbers of

hidden nodes in different numbers of hidden layers. The model built with $28 \times 38 \times 19$ hidden nodes performed the best, and had an $R^2$ of 0.843 and an MSE of 0.367 for Adj3 (like–dislike), which was superior to the conventional ANN model. Increasing the number of hidden nodes and hidden layers in conventional ANN does not typically improve the MSE and $R^2$. Table 7 further reveals that the training of the ANN was relatively satisfactory when three hidden layers were employed; however, the training was less favorable and the training time was increased when four hidden layers were employed. Thus, three were selected as the most appropriate number of hidden layers.

**Table 7.** Predictive performance of different GA-based ANN models.

| Models | Hidden Layers | Hidden Nodes | MSE | | | $R^2$ | | | Overall $R^2$ |
|--------|---------------|--------------|------|------|------|------|------|------|------|
| | | | Adj1 | Adj2 | Adj3 | Adj1 | Adj2 | Adj3 | |
| ANN | 2 | $10 \times 10$ | 0.549 | 0.308 | 0.536 | 0.536 | 0.555 | 0.757 | 0.627 |
| ANN | 2 | $50 \times 50$ | 0.524 | 0.259 | 0.469 | 0.550 | 0.619 | 0.782 | 0.662 |
| ANN | 2 | $100 \times 50$ | 0.690 | 0.340 | 0.552 | 0.441 | 0.496 | 0.755 | 0.569 |
| ANN | 2 | $50 \times 100$ | 0.828 | 0.417 | 0.967 | 0.34 | 0.4 | 0.61 | 0.357 |
| ANN | 3 | $50 \times 50 \times 50$ | 0.480 | 0.312 | 0.531 | 0.61 | 0.51 | 0.72 | 0.624 |
| GA-ANN | 1 | 16 | 0.419 | 0.231 | 0.403 | 0.633 | 0.665 | 0.824 | 0.713 |
| GA-ANN | 2 | $35 \times 5$ | 0.393 | 0.222 | 0.417 | 0.652 | 0.683 | 0.829 | 0.724 |
| GA-ANN | 3 | $28 \times 38 \times 19$ | 0.381 | 0.239 | **0.367** | 0.668 | 0.674 | **0.843** | **0.733** |
| GA-ANN | 4 | $30 \times 46 \times 30 \times 46$ | **0.360** | **0.209** | 0.472 | **0.699** | **0.703** | 0.806 | 0.720 |

Adj1 (elegant–artless); Adj2 (rare–common); Adj3 (like–dislike). Bold numbers indicates the optimal value.

## 5. Discussion

This section presents the results of the GA-ANN modeling using scatter plots. Figure 5 displays the detailed results of models with 1–3 training layers. Figure 5a presents the training results of the model with one hidden layer. The hidden layer contained 16 nodes. The $R^2$ of the adjective sets were 0.633, 0.665 and 0.824; these results are superior to those obtained with an arbitrarily selected ANN structure. Figure 5b reveals that when the model comprised two hidden layers and $35 \times 5$ nodes, the $R^2$ of the adjective sets were 0.652, 0.683 and 0.829, respectively, whereas when the model comprised three hidden layers and $28 \times 38 \times 19$ nodes, the $R^2$ of the adjective sets were 0.668, 0.674 and 0.843 (Figure 5c). The data distribution was relatively highly concentrated, suggesting that the GA-ANN under these conditions is the optimal model. In particular, the highest correct prediction rate was found for Adj3 (like–dislike), which indicates respondents' preference.

Tables 8 and 9 present the shoe color schemes and types that resulted in relatively small and large prediction errors using the GA-ANN, respectively. These prediction data were compared with statistics obtained from the original questionnaire responses, which revealed that for shoe types for which the predictions were accurate, the questionnaire investigation results were comparably coherent, and the trends were identifiable. A review of these samples suggested that the prediction results for casual and running shoes were the most accurate, whereas the prediction results for shoes with a shape that is relatively difficult to identify were the least accurate. Regarding shoes with two colors, the prediction accuracy for shoes with black soles was relatively higher than for single-color shoes. Therefore, the effect of color could not be predicted; the prediction accuracy for different colors varied randomly. This was found by plotting the colors with high and low prediction accuracies on the color wheel, in which no identifiable trend could be observed. This study inferred that the respondents had relatively similar opinions of the different shoe shapes, so trends in their perceptions were easily identified. By contrast, the opinions of the respondents regarding the color schemes were different, so no obvious trend could be identified.

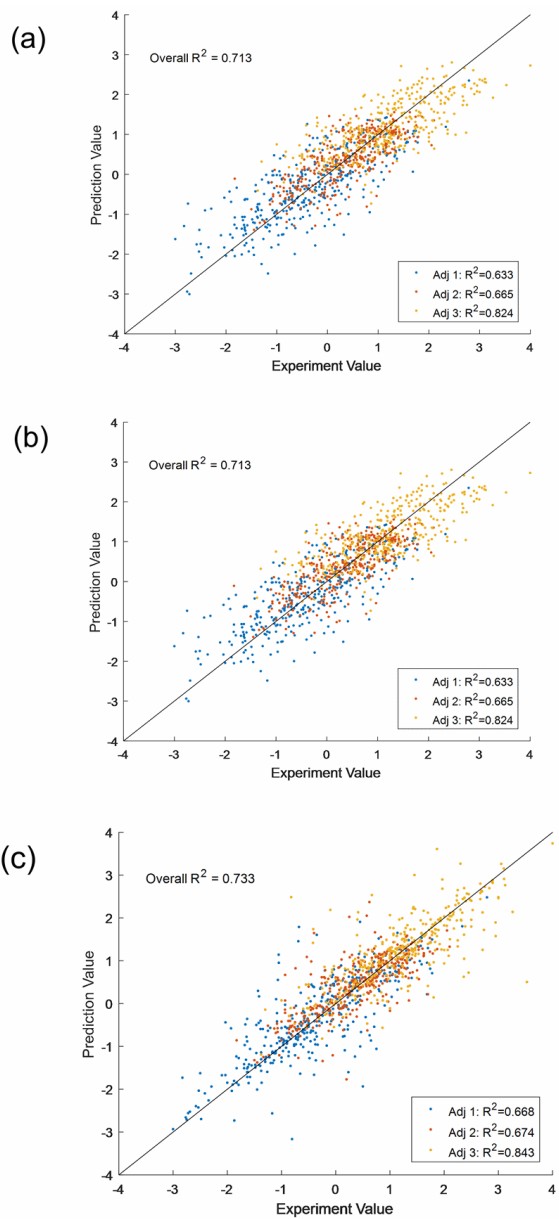

**Figure 5.** Prediction results of models with different numbers of hidden layers. (**a**) one hidden layer (16); (**b**) two hidden layers (35 × 5); (**c**) three hidden layers (28 × 38 × 19).

In addition to th econfigurations of solid colors (color = hue), this study included gradient color effects, because these more closely reflect product designs currently on the market. Figure 6 presents the curves representing the difference between the predictions for five experiment levels, employing gradient colors using the GA-ANN and actual questionnaire statistics. The three curves in Figure 6 separately represent the results for the three adjective pairs. The dotted lines indicate GA-ANN predictions, whereas the dots indicate actual experimental values. The closer the dots to the curves, the more accurate the prediction results are. Whereas Figure 6 shows the results for horizontal color gradient effects, Figure 7 depicts those for vertical color gradient effects. A comparison of these two figures reveals that the GANN predictions were slightly superior when horizontal color gradients were used. This indicates that if designers wish to use a color gradient effect to stimulate consumers, they should employ a horizontal gradient (Figure 6), because Kansei design determines that it is more effective at triggering consumer feelings.

**Table 8.** Close-to-optimal parameters generated by GA optimization on the trained ANN model.

| Shoe Form Type | Color Quantity | Gradient or Color Index | Optimal Shoecolor Schematic | |
| --- | --- | --- | --- | --- |
| | | | Opt. Value | Questionnaire Average Score |
|  | 2 | Horizontal | −0.05, 0.25, 1.45 | −0.12, 0.39, 1.43 |
|  | 2 | 2<br>229.0.79 | −0.69, 0.86, −0.07 | −0.72, 0.47, 0.73 |
|  | 2 | Vertical | −0.57, 1.03, 1.13 | −0.43, 0.96, 1.27 |
|  | 2 | 16<br>0.155.107 | −0.60, 0.17, 0.80 | −0.17, 0.04, 0.99 |
|  | 2 | 8<br>29.32.136 | −0.46, −0.54, 1.43 | −0.88, −0.57, 1.21 |
|  | 2 | 15<br>0.158.150 | −0.46, −0.43, 0.82 | −0.67, −0.11, 0.76 |
|  | N | Horizontal | −1.92, −0.81, 0.81 | −0.24, −0.75, 0.76 |
|  | 2 | 15<br>0.158.150 | −0.03, 0.30, 0.97 | 0.16, 0.54, 0.94 |
|  | 2 | 4<br>228.0.127 | 0.71, 0.61, 1.48 | −0.54, 1.68, 0.37 |

**Table 9.** Shoe designs for which relatively large prediction errors in the ANN were generated.

| Shoe Form Type | Color Quantity | Gradient or Color Index | Optimal Shoe Color Schematic | |
|---|---|---|---|---|
| | | | Opt. Value | Questionnaire Average Score |
|  | 1 | 17<br>0.158.68 | 0.71, 1.29, 1.45 | 1.71, 1.14, 3.00 |
|  | 1 | 22<br>252.200.0 | −0.60, −0.07, 1.87 | −0.60, 3.44, 0.61 |
|  | 1 | 8<br>29.32.136 | 2.33, 1.75, 2.17 | 1.81, 1.43, 2.13 |
|  | 1 | 16<br>0.155.107 | −0.60, 0.17, 0.80 | −0.17, 0.04, 0.99 |
|  | 1 | 6<br>146.7.131 | −0.46, −0.54, 1.43 | −0.88, −0.57, 1.21 |
|  | 1 | 21<br>29.32.136 | −1.05, 0.40, 1.00 | 0.95, 1.65, 2.07 |
|  | 1 | Horizontal | −1.17, 0.24, 0.86 | −2.56, −1.34, -0.56 |
|  | 1 | Horizontal | −0.29, 0.57, 1.43 | 0.35, 0.25, 1.22 |
|  | 1 | Horizontal | 0.45, 0.62, 0.41 | 1.91, 2.37, 2.13 |

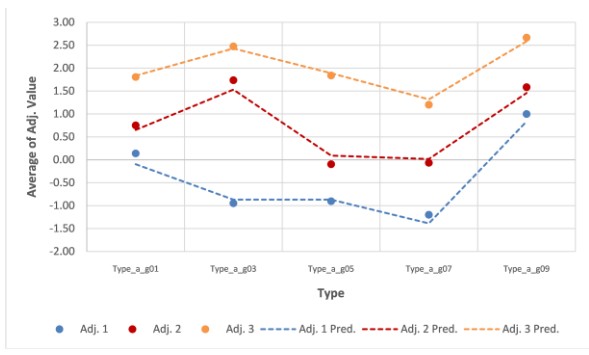

**Figure 6.** Prediction performance for horizontal color gradients.

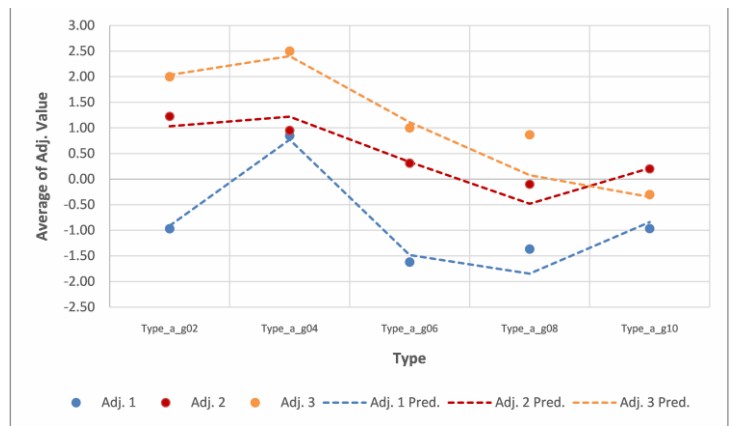

**Figure 7.** Prediction performance for vertical color gradients. Adj1 (elegant–artless); Adj2 (rare–common); Adj3 (like–dislike).

## 6. Conclusions

The design factors of the sports shoes in this study were the input parameters of the ANN, and GA was used to search for the optimal ANN structure. It was found that the GA-based ANN model outperformed the conventional ANN model. The results indicated that a model built with three hidden layers [28 × 38 × 19] was best for predicting the object value reliability. The $R^2$ for Adj3 (like–dislike) was equal to 0.834, suggesting that the developed model is a feasible and efficient tool for predicting the objective value of product images. Additionally, comparing the optimized values and questionnaire average values demonstrated that the model's prediction accuracy was higher for shoe bodies with two colors than those with a single color. Highly accurate prediction results were obtained for casual and running shoes, whereas shoes that had a shape that was relatively difficult for respondents to identify returned inaccurate predictions. The colors of samples that resulted in higher prediction rates were random (i.e., no specific pattern was identified), suggesting that the respondents had varying personal color preferences.

The highest accuracy rate was obtained for Adj3 (like–dislike), indicating that composite models, such as GA-ANN, can facilitate the prediction of consumer preference for products under development to a certain extent.

In this paper, we have presented a hybrid GA-ANN predictive model for searching close-to-optimal sports shoe color combinations for a given product. The ANN was used to determine consumer perceptions for 360 experimental samples according to the concept of Kansei engineering. However, further research identifying any correlations or interactions among respondents' preferences for particular color combinations, predicting the proper shoe brand and size, and implementing specific testing protocols for footwear fit and comfort perception, is warranted. Although the current study

focused on the color scheme design of sports shoes, the proposed method is suitable for other design issues in the development of footwear and related products.

**Funding:** This research received no external funding.

**Conflicts of Interest:** The authors declare no conflict of interest.

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
