# Peer review of "Prediction of Optimized Color Design for Sports Shoes Using an Artificial Neural Network and Genetic Algorithm"

_applsci, doi:10.3390/app10051560_

Round 1

Reviewer 1 Report

Major point: The author is trying to persuade us that the GA-ANN training method is better than a simple ANN training regime (e.g. via backpropagation) but the proposed comparison is not fair for the ANN. In order for this comparison to be fair the capacity of the comparing architectures (i.e. the total amount of weights) should be of equal magnitude. Here we have a 10+10 ANN vs a 30x46x30x46 GA-ANN. A more fair architecture for the NN should have been a 50+50 or (even better) a 100+50. There are also recommendations on how to select a fixed amount of neurons for the architecture of an ANN that have not been taken into account.   Minor point: There is no mention of other key parameters of the ANN architecture: what are the activation functions of the hidden layers (eg tanh, ReLU, etc) and the output? Are there biases?  What type of training was conducted by the ANN   Minor point: The author has selected four types of currently available sports shoes from the current market and the final type of shoe was selected from the three-dimensional (3D) model resource and was unmodified by product design considerations.  However, two out of the final five 3D designs, have a distinguished mark that makes them come out of the other 3 designs; the three stripes are the trademark of well known sportswear company. This makes the specific shoes to stand out when the participants evaluate the final coloured design. This human bias should have been considered in the proposed training method, should also have been acutely aware of those risks and working to reduce them. 

Author Response

Responses to the comments of Reviewer #1

  1. The author is trying to persuade us that the GA-ANN training method is better than a simple ANN training regime (e.g. via backpropagation) but the proposed comparison is not fair for the ANN. In order for this comparison to be fair the capacity of the comparing architectures (i.e. the total amount of weights) should be of equal magnitude. Here we have a 10+10 ANN vs a 30x46x30x46 GA-ANN. A more fair architecture for the NN should have been a 50+50 or (even better) a 100+50. There are also recommendations on how to select a fixed amount of neurons for the architecture of an ANN that have not been taken into account.  

Response:

Thanks for your suggestion. We modified Table 7 (Predictive performance of different GA-based ANN models) on page 13 by adding more ANN training results. The results demonstrated better performance when the number of hidden nodes and hidden layers was increased in conventional ANNs, which resulted in the higher performance of GA-ANNs.

  1. There is no mention of other key parameters of the ANN architecture: what are the activation functions of the hidden layers (eg tanh, ReLU, etc) and the output? Are there biases? What type of training was conducted by the ANN

Response:

Thanks for your suggestion. The default type of transfer function was selected for the hidden and output layers in MATLAB—hyperbolic tangent sigmoid transfer function (tansig) for the hidden layer and log-sigmoid transfer function (logsig) for the output layer. Backpropagation was used to train the ANNs. Section 4.2 (Description of the ANN Model) on page10 provides the related description.

  1. The author has selected four types of currently available sports shoes from the current market and the final type of shoe was selected from the three-dimensional (3D) model resource and was unmodified by product design considerations. However, two out of the final five 3D designs, have a distinguished mark that makes them come out of the other 3 designs; the three stripes are the trademark of well known sportswear company. This makes the specific shoes to stand out when the participants evaluate the final coloured design. This human bias should have been considered in the proposed training method, should also have been acutely aware of those risks and working to reduce them. 

Response:

Thank you for the indication. This was an error that occurred when building the appropriate 3D sample shoe; this was conducted to prevent participants from experiencing experiment fatigue or any other learning effects; this will be considered in future studies .

Reviewer 2 Report

The paper presents a hybrid method (ANN + GA) to identify the best colour scheme to use for sports shoes. The research theme is coherent with the typical trends of the footwear sector. The method presented is innovative enough to deserve to be published.

The title correctly reflects the goal of the paper. The abstract summarizes the content of the paper correctly. The keywords are consistent.

Introduction and state of the art are presented in great detail. However, these two chapters are very extensive in relation to the others in the paper.

The state of the art is lacking in articles related to the footwear sector. There are many scientific papers in which artificial intelligence methods are used to support the design and configuration of footwear (e.g.; Shan Huang, Zhi Wang, Yong Jiang, Guess your size: A hybrid model for footwear size recommendation, Advanced Engineering Informatics, Volume 36, April 2018, Pages 64-75). It would be useful to take this into consideration.

Although the ratings have a scale from -10 to +10, scatter plots highlight how user preferences are generally in the -4 + 4 range. Is there a particular reason?

Chapter 4 should be renamed to Results, considering that Chapter 5 is related to discussion.

Future developments should be further detailed in relation to the results obtained in this paper.

Author Response

Responses to the comments of Reviewer #2

  1. The state of the art is lacking in articles related to the footwear sector. There are many scientific papers in which artificial intelligence methods are used to support the design and configuration of footwear (e.g.; Shan Huang, Zhi Wang, Yong Jiang, Guess your size: A hybrid model for footwear size recommendation, Advanced Engineering Informatics, Volume 36, April 2018, Pages 64-75). It would be useful to take this into consideration.

Response:

We have added several new references related to shoe design research to Section 1; these include the following:

  1. Huang, S.; Wang, Z; Jiang, Y. Guess your size: A hybrid model for footwear size recommendation. Adv. Eng. Inform. 2018, 36, 64–75
  2. Lee, Y.C.; Wang, M.J. Taiwanese adult foot shape classification using 3d scanning data. Ergonomics. 2015, 58, 513–523.
  3. Alcantara, E.; Artacho, M.A.; Gonzalez, J.C.; Garcia, A.C. Application of product semantics to footwear design. Part I—Identification of footwearsemantic space applying differential semantics. Int. J. Ind. Ergonom. 2005, 35, 713–725.
  4. Chang, C.A.; Lin, M.C.; Leonard, M.S.; Occena, L.G. Building an expert system for the design of sports shoes. Comput. Ind. Eng. 1988, 15, 72–77.
  5. Sudta, P.; Kanchan, K.; Chantrapornchai, C. Children shoes suggestion system using data mining. Int. J. Database Theory Appl. 2012, 5, 21–36.
  6.  
  1. Although the ratings have a scale from -10 to +10, scatter plots highlight how user preferences are generally in the -4 + 4 range. Is there a particular reason?

Response:

Here, the scoring scale was polarized (e.g., the highest scores for modern and retro were 210 and 10, respectively), and user preferences were generally in the −4 to +4 range (i.e., relatively low). Based on the relative differences in these values, referential values were retained for objective quantification. Thus, in future studies, a rating scale in the −5 to +5 range should be used to obtain an obvious scatter result to better understand the context of the paper.

  1. Chapter 4 should be renamed to Results, considering that Chapter 5 is related to discussion. Future developments should be further detailed in relation to the results obtained in this paper.

Response:

We have renumbered Section 4 to Section 5 and have added some future study suggestion at the end of Section 6

Reviewer 3 Report

  The paper is very interesting and deals with a topic of great importance for industrial design. Designers are often influenced by their own interests. Using metaheuristics and optimization methods such as genetic algorithm and neural networks, combined with Kansei engineering (to know the user-product interaction) is really interesting.   Congratulations to the authors.
As comments to improve the paper:
- Improve section 2 (literatutal review) by introducing other examples where kansei engineering is combined with metaheuristics.
- Improve section 2 (literatutal review) with studies on product design where ANN and GA are applied.
- Expand the section on the state of the art (sections 2.1, 2.2 and 2.3) on ANN and GA, so that the reader can better understand the context of the paper (and the methods).
- What are the future works in this line of study?

Author Response

Responses to the comments of Reviewer #3

  1. - Improve section 2 (literatutal review) by introducing other examples where kansei engineering is combined with metaheuristics.

Response:

We have added to Section 2.1 (Kansei Engineering) several research examples where Kansei engineering was combined with multiple regression analysis (Han & Hong, 2003), quantification theory type 1 (Smith & Fu, 2011), fuzzy theory logic (Yang, 2007), rough set theory (Shieh, 2016), procrustes analysis (Wang, 2015), genetic algorithms (GAs) (Lin, 2014), and neural networks (NNs) (Tsai, 2005).

  1. - Improve section 2 (literatutal review) with studies on product design where ANN and GA are applied.

Response:

We have added citations of several studies presenting how GA could be introduced in the field of design: Beale constructed a data mining database that uses GAs to measure the correlation between web pages and user interests, Hsiao demonstrated the effectiveness of GAs for assessing coffee maker design feasibility [35], and Wang employed the interactive GAs and the fuzzy kano model to explore the emotional needs of users for electric bicycle design.

  1. - What are the future works in this line of study?

Response:

We have renumbered Section 4 to Section 5 and have added some future study suggestions at the end of Section 6.

Round 2

Reviewer 1 Report

Changes according to the comments were followed. The v2 of the paper is certainly improved comparing to the 1st one.